# Persistence of *Campylobacter* spp. in Poultry Flocks after Disinfection, Virulence, and Antimicrobial Resistance Traits of Recovered Isolates

**DOI:** 10.3390/antibiotics12050890

**Published:** 2023-05-10

**Authors:** Manel Gharbi, Awatef Béjaoui, Safa Hamrouni, Amel Arfaoui, Abderrazak Maaroufi

**Affiliations:** Group of Bacteriology and Biotechnology Development, Laboratory of Epidemiology and Veterinary Microbiology, Institut Pasteur de Tunis, University of Tunis El Manar (UTM), Tunis 1002, Tunisia; awatef.bejaoui@pasteur.tn (A.B.); safa.hamrouni@pasteur.tn (S.H.); abderrazak.maaroufi@pasteur.tn (A.M.)

**Keywords:** campylobacter, antibiotic resistance, resistance mechanisms, virulence, breeding hens, environment, risk factors

## Abstract

To investigate the persistence risk of *Campylobacter* spp. in poultry farms, and to study the virulence and antimicrobial resistance characteristics in the recovered strains, we collected 362 samples from breeding hen flocks, before and after disinfection. The virulence factors were investigated by targeting the genes; *flaA*, *cadF*, *racR*, *virB11*, *pldA*, *dnaJ*, *cdtA*, *cdtB*, *cdtC*, *ciaB*, *wlaN*, *cgtB*, and *ceuE* by PCR. Antimicrobial susceptibility was tested and genes encoding antibiotic resistance were investigated by PCR and MAMA-PCR. Among the analyzed samples, 167 (46.13%) were positive for *Campylobacter*. They were detected in 38.7% (38/98) and 3% (3/98) of environment samples before and after disinfection, respectively, and in 126 (75.9%) out of 166 feces samples. In total, 78 *C. jejuni* and 89 *C. coli* isolates were identified and further studied. All isolates were resistant to macrolids, tetracycline, quinolones, and chloramphenicol. However, lower rates were observed for beta-lactams [ampicillin (62.87%), amoxicillin-clavulanic acid (47.3%)] and gentamicin (0.6%). The *tet(O)* and the *cmeB* genes were detected in 90% of resistant isolates. The *bla*_OXA-61_ gene and the specific mutations in the *23S rRNA* were detected in 87% and 73.5% of isolates, respectively. The A2075G and the Thr-86-Ile mutations were detected in 85% and 73.5% of macrolide and quinolone-resistant isolates, respectively. All isolates carried the *flaA, cadF, CiaB, cdtA, cdtB,* and *cdtC* genes. The *virB11*, *pldA,* and *racR* genes were frequent in both *C. jejuni* (89%, 89%, and 90%, respectively) and *C. coli* (89%, 84%, and 90%). Our findings highlight the high occurrence of *Campylobacter* strains exhibiting antimicrobial resistance with potential virulence traits in the avian environment. Thus, the improvement of biosecurity measures in poultry farms is essential to control bacterial infection persistence and to prevent the spread of virulent and resistant strains.

## 1. Introduction

*Campylobacter* spp. are a major cause of foodborne diarrheal illnesses in humans, and represent the main cause of infant enteric disease in developing countries. Campylobacteriosis is an important zoonosis, with infection occurring mainly through the ingestion of contaminated food and water, or direct contact with contaminated food animal species or their carcasses [1]. Out of the 17 species and the 6 subspecies of the genus *Campylobacter* spp., *C. jejuni* and *C. coli* are the most commonly documented species in human diseases. The infection causes inflammation, bloody diarrhea, cramps, fever, and pain. Even though the pathogenesis of *Campylobacter* spp. is yet unclear, various virulence factors, such as motility, adhesion, colonization, host cell invasion, and the production of cytolethal distending toxin were associated with their pathogenicity [1,2].

For a human *Campylobacter* infection, supportive antimicrobial therapies are typically not necessary; however, in the case of immunocompromised people, pregnant women, children, and the elderly, an antibiotic therapy could be needed. Fluoroquinolones (i.e., ciprofloxacin) and macrolides (erythromycin, azithromycin) are the most prescribed drugs for human campylobacteriosis; in addition, tetracycline and gentamicin are also effective against systemic infection with *Campylobacter* spp. [3]. These common antibiotics are frequently used in food animals, such as chicken, to prevent and reduce bacterial infections in farms and to improve the performance of the animals’ growth. Globally, these practices have been associated with high rates of antimicrobial resistance (AMR) in *Campylobacter* spp. isolated from animal sources. The loss of therapeutic effectiveness of antibiotics due to AMR spreading leads to higher rates of mortality and morbidity from infectious diseases both in humans and animals, generating serious socioeconomic and public health problems [3].

*Campylobacter* spp. infect several animals (cattle, sheep, pigs, birds, reptiles, and crustaceans, etc.); however, poultry are the main reservoir of these agents. The bacteria colonize the cecum, distal jejunum, and cloaca of birds, given their ability to live in the mucus and crypts of these organs. Most of *Campylobacter* species colonize and proliferate in the chicken gastrointestinal tract without any clinical symptoms [4,5]. They can survive and spread from one rearing cycle to the next despite their tremendous vulnerability in the breading farm. In addition, since chickens are coprophagic, inadequate biosecurity measures and an intensive production method are the main causes of infection spreading from infected chicken to others [6]. Consequently, fecal shedding of *Campylobacter* allows this pathogen to easily contaminate the carcasses during processing, which can ultimately lead to the transmission of *Campylobacter* to humans.

*Campylobacter* could also be introduced into flocks from the immediate environment of the buildings as well as from surrounding farms [7,8] by various vectors: insects, rodents, farmers or other persons entering the buildings during rearing, as well as equipment introduced from one building to another [9,10]. The litter also plays an important role as a source for contamination, indeed, these bacteria could survive for 10 days at 20 °C [11].

In Tunisia, as in other developing countries, data on the status of *Campylobacter* in poultry farms are limited. Today, it has become essential to take the problem of farm contamination into consideration, both for its impact on public health and for the significant economic repercussions for farmers. Although contamination of poultry meat is possible at all stages of the food chain, the rearing period represents a critical phase in the spreading of the bacteria. Therefore, the aim of this study was to investigate the persistence of *Campylobacter* spp. contamination in one Tunisian intensive breeding hen farm after cleaning and disinfection and to characterize virulence and antimicrobial resistance traits of the recovered *Campylobacter* strains.

## 2. Results

### 2.1. Biosecurity Measures

#### 2.1.1. Environment of the Farm

In the environment of the visited farm, animals (cats and dogs), insects and reptiles, also known as vermin, were present. The studied farm is located near the villages that surround it, and close to traffic routes. The buildings (*n* = 28) are more than 20 m apart from each other. The areas around the buildings are paved, and the parking areas for delivery vehicles are always close to the building’s entrance (Appendix A).

#### 2.1.2. Characteristics of the Farm

The capacity of the breeding hen flocks varied between 2500 to 18,000 birds in each building, and the average surfaces varied from 500 to 1600 m^2^. The age of the sampled hens ranged from 42 days to 75 weeks. The floors were made of concrete and the litter was made of wood chips in all of the buildings. The ventilation system was automatic. The buildings contained an airlock separated into two zones with a pedestrian restroom at the entrance. The farm managers declared that they clean the airlock floor at least once a week (Appendix A).

#### 2.1.3. Sanitary Measures

The application of sanitary measures for environmental cleaning is done on a regular basis. The farm is distinguished by a high animal density per m^2^. In fact, at the time of installation (hens), the density in the breeding hen flocks ranged between 10 and 30 animal/m^2^. Water was supplied to the visited farm by the National Society for the Exploitation and Distribution of Water (SONED) network. Employees of the farm clean and disinfect the facilities. A sanitary vacuum is used, with a duration of 20 days.

#### 2.1.4. Personnel Hygiene 

The number of workers per building does not exceed two. A change of clothes and boots was not strictly respected (at least once a week).

According to the biosecurity data recovered from the questionnaire, the most applied biosecurity measures in this farm were locked buildings, visitor restriction, feed quality, united overalls and footwear for the workers, and the application of a disinfection program at the end of each rearing cycle. By contrast, the lowest applied measures were the high intensity of birds per m^2^, the isolation of the farm from animals (rodents, reptiles, dogs, cats, etc.), absence of disinfectant baths, low clothes property, and hand hygiene.

### 2.2. Contamination Rates of Campylobacter spp. in Breeding Hens and Environment before and after Cleaning and Disinfection

The rate of *Campylobacter* detection in the cloacal swabs of breeding hens was 75.9% (126/166); 70.63% of isolates were identified as *C. jejuni* (*n* = 89) and 29.36% as *C. coli* (*n* = 37). The percentage of infection in the flocks ranged from 14% to 56% per building (*p* < 001). 

A total of 196 samples were collected from the environment of eight breeding hen buildings before and after cleaning and disinfection (Table 1). Before the void spaces, we collected 98 samples, from plates, nest boxes, and soil. Among them, 38 (38.7%) were *Campylobacter* spp.-positive, which were identified as *C. coli*.

Among the 98 environmental samples collected after cleaning and disinfection, only three boots were *Campylobacter* spp.-positive (3%; 3/98) (Table 1). The three were identified as *C. coli*. Taken together, 89 *C. jejuni* and 78 *C. coli* isolates were identified and further studied.

### 2.3. Antimicrobial Resistance Patterns

All *Campylobacter* isolates were resistant to erythromycin, ciprofloxacin, nalidixic acid, chloramphenicol, and tetracycline. To a lesser extent, resistance rates to β-lactams were about 62.87% toward ampicillin and 47.3% toward amoxicillin/clavulanic acid. The resistance rates in *C. jejuni* and *C. coli* isolates against ampicillin (47.19% and 81%, respectively) and amoxicillin/clavulanic acid (58.42% and 20.22%, respectively) differed significantly (*p* < 0.05). No significant difference was observed between both species for the other antibiotics (Table 2).

Multidrug resistance to at least three antimicrobial classes was detected in all *Campylobacter* isolates, and 14.37%, 47.9%, and 30.53% of isolates were resistant to 4, 5, and 6 antimicrobial classes, respectively. When looking at the antimicrobial resistance patterns, five antimicrobial resistance patterns were found for all *Campylobacter* isolates, with the combination “AMP + CIP + NAL + ERY + TET + CHL” as the most common profile (39.52%). A significant higher percentage of *C. coli* isolates (83.3%) exhibited this resistance profile, compared with *C. jejuni* isolates (47.6%). The next most frequent pattern was the combined resistance to AMP, AMC, CIP, NAL, ERY, TET, and CHL, detected in 30.53% of isolates. The remaining patterns comprised less than 10% of isolates (Figure 1).

### 2.4. Detection of Antimicrobial Resistance Genes

The rates of antimicrobial resistance genes among resistant isolates of *C. jejuni* and *C. coli* were as follows: *tet(O)* (100% vs. 80%), *cmeB* (80% vs. 100%), and *bla_OXA-61_* (81% vs. 93%), respectively. The *aphA-3* gene was not detected (Figure 1). Interestingly, when testing these genes in susceptible isolates, the *bla*_OXA-61_ gene was detected in 41% of β-lactams-susceptible *C. coli* isolates. None of the other genes was detected in the sensitive isolates. In Cip and/or Nal resistant isolates, the analysis of the *gyrA* gene showed the presence of the Thr-86-Ile amino acid substitution in 90% of *C. jejuni* and 80% of *C. coli* isolates. Meanwhile, for the only *C. jejuni* isolate exhibiting susceptibility to both quinolones (Cip and Nal), the Thr-86-Ile substitution was not detected. Macrolide-resistant isolates were tested for the presence of the mutations A2074C and A2075G in the *23S rRNA* gene. The mutation at position A2075G was found in 86% and 61% of *C. jejuni* and of *C. coli* isolates, respectively. While the A2074C mutation occurred in 14% and 27% of *C. jejuni* and *C. coli* isolates, respectively. Interestingly, 12% of *C. coli* isolates harbored mutations at both positions (A2074/2075) of the V region (Figure 2).

### 2.5. Prevalence of Genes Encodingvirulence Factors

All *Campylobacter* isolates (78 *C. jejuni* and 89 *C. coli*) were analyzed for virulotype (content of genes coding for virulence factors) and AR phenotypic profiles. All isolates (*n* = 167, 100%) harbored the *flaA, cadF, ciaB*, and *cdt* genes, followed closely by the *racR* gene (*n* = 160, 90%). A similar result was obtained when analyzing the 78 *C. jejuni* isolates. In addition, the *flaA, cadF, ciaB*, and *cdt* genes were present in all *C. jejuni* isolates (*n* = 78, 100%), followed by the *dnaJ* (*n* =73, 93.58%) and *ceuE* (*n* = 69, 88.46%) genes (Figure 3A). No discernible differences were found between both campylobacter species for the most prevalent virulence genes. Indeed, all isolates contained the *flaA, cadF, racR, ciaB*, and *cdt* genes, while the *pldA* gene was detected in 75 (84.26%) isolates. Interestingly, a significant difference was found for the *ceuE* gene, which was absent in all *C. coli* isolates but strongly present in *C. jejuni* (90.47%) (Figure 3B).

## 3. Discussion

### 3.1. Prevalence of Campylobacter and Risk Factors

In this study, we focused on the risk factors contributing to the colonization of *Campylobacter* in breeding hens, in order to evaluate the control measures to be adopted to reduce the prevalence of *Campylobacter* colonization. The described factors are related to the farm management and the sanitary measures applied in the flocks. The survey in this poultry farm showed that several animals such as cats, dogs, insects, and reptiles have access to these facilities. It is well known that these animals can carry zoonotic pathogens [4].

The proximity of farms to rural villages is also a problem for the poultry sector in some cases. This results in the negative influence of certain human activities on the studied farm. The cohabitation of humans with these animals may carry the risk of increasing the possibility of human contamination [12]. The absence of an outer barrier on the farm gives free access of outside animals such as dogs and cats. This lack of a barrier at the farm level can be a factor in the degradation of hygiene at this farm. As a result, it might have a negative effect on productivity and possibly act as a vector for the spread of some zoonotic diseases to poultry [13]. Easy access to poultry facilities by dogs, cats, and reptiles significantly increases the risk of contamination. Those studies also indicated that any input to the farm is likely to carry the bacteria from infected units to others, as is the case for personnel in the absence of single-use boot swabs or equipment soiled by feces [12,14]. The same applies to vehicles driving around the building.

The cleaning of farms is an effective way to reduce and even eliminate certain pathogenic microorganisms. The studied farm consisted of 28 flocks, which were made of concrete floors and were easy to clean and disinfect and therefore not favorable to the survival and multiplication of pathogenic bacteria.

Our work showed that the poultry farm exhibited a density of poultry that was above regulatory levels, which is in accordance with the results of other authors who have reported that high density is a risk factor for contamination and facilitates the growth of certain pathogenic bacteria [15]. Furthermore, previous studies have also shown that crowding is a key factor in the introduction of pathogens into an environment [16]. Knowledge of these practices can help to reduce the risks in advance. In fact, Jussiau et al. [17] showed that when personnel perform their duties without consistently changing into new work clothing at least once a week and without using single-use boot covers, they promote the contamination of the farms by pathogenic *Campylobacter* spp. 

The studied poultry production structure has a production system showing average health safety measures. The risk factors for *Campylobacter* infection in poultry products are real and are summarized, for example, by the precarious nature of the installations and the non-respect of hygiene rules by certain operators in the sector. Thus, the studied farm could be a potential source of dissemination of zoonotic infections mainly campylobacteriosis.

In the domain of food safety, *Campylobacter* represents an emerging threat with increasing importance over the last years. Therefore, in our work, we were interested in studying the prevalence of *Campylobacter* spp. in the feces of breeding hens. In the present study, our findings revealed a prevalence of 75.9% (126/166 samples) from which 70.6% and 29.36% were identified as *C. jejuni* and *C. coli*, respectively. *C. jejuni* was the predominant species recovered from poultry carcasses as reported in several studies worldwide [18]. When comparing the *Campylobacter* infection rate in the studied farm (39.4%), it showed that it was higher than rates observed in Greece (13.33%) [19] and in Australia (11%) [20], but it is lower than those observed in Finland (86%) [21], Italy (65%) [22], and Sri-Lanka (64%) [23].

Despite the various epidemiological studies carried out on *Campylobacter* contamination of farms, the sources and routes of contamination of chickens by this bacterium are still poorly understood. It is essential to consider the problem of farm contamination, both for its impact on public health and for the significant economic repercussions. Although contamination of meat is possible at all levels of the production chain, the rearing period is also a critical stage for the development of *Campylobacter* [24]. The knowledge of the modalities of contamination of poultry by *Campylobacter* during this period is therefore essential in order to prevent its development throughout the chain. To our knowledge, no study has been done in Tunisia on the factors of *Campylobacter* contamination in breeding hens’ flocks. Therefore, our objective in this study was to investigate the prevalence of *Campylobacter* in the environment of a poultry farm and to determine the possibility of isolating *Campylobacter* from environmental surfaces, flocks, and changing rooms after cleaning and disinfection procedures in flocks. Our study was conducted between May and June 2018, when the weather became dry, as it has been previously reported that *Campylobacter* survival was reduced in low humidity environments [25,26].

A total of 196 samples were collected from the farm environment before and after cleaning and disinfection of the flocks. Before and after the sanitation, we collected 98 environmental samples. The overall prevalence of *Campylobacter* spp., was 21% (41 isolates) all identified as *C. coli*. The predominance of *C. coli* in isolates from breeding hen flocks environments was in agreement with various surveys conducted in European countries, including Bulgaria, Hungary, Italy, Luxembourg, Malta, Portugal, Ireland, Spain, and Greece [27,28,29,30]. Of all the samples taken before cleaning, 38.7% (38/98) were positive for *C. coli*. The isolation rate in our study is in line with previous results in France [31,32]. From the total number of samples taken after cleaning, only 3% (3/98) were positive for *C. coli* which were isolated from the workers’ boots. The results indicate that transmission to the chickens can occur via the farmer (boots).

It has also been shown that the implementation of strict biosecurity measures (such as boot dips) has an effect on the prevalence of *Campylobacter*, reducing the transmission rate [33]. This observation corroborates with the results of Guerin et al. [33] and Facciolà et al. [34] showing the presence of *Campylobacter* in the immediate environment of farms. 

Our findings are also explained by insufficient environmental conditions for cleaning and disinfection, poor hygiene conditions and ignorance of owners and workers. The persistence of *Campylobacter* after cleaning and disinfection of poultry farms is commonly reported in poultry industries and constitute a serious threat for human health [35,36]. To summarize, although poultry farming is evolving in Tunisia, there are still shortcomings in the management of the quality of the facilities and the application of good practices for the rearing and slaughter of poultry products. 

### 3.2. Antimicrobial Resistances and Implicated Genes

The treatment of infections in humans and animals with antibiotics is being threatened by the emergence and spread of antimicrobial resistant bacteria. Inappropriate use of veterinary drugs in animal husbandry plays an important role in the rise of antimicrobial resistance rates in foodborne pathogens. Antibiotics are extremely used in high-volume in poultry production, whether to prevent bacterial infections or as growth promoter. Thus, antimicrobial resistance studies are essential to identify resistant *Campylobacter* strains in poultry and their environments. The spread of antimicrobial resistance worldwide is strongly linked to mobile genetic elements like plasmids and transposons, which can also carry virulence determinants [37].

Our research has shown that resistance to erythromycin, tetracycline, quinolones and ciprofloxacin is prevalent, which can significantly reduce the possibilities of treating infections caused by these strains. As these antibiotics have been on the market for a long time and have been frequently used in legal and illegal situations, high rates of resistance are highly expected. Similar high rates of resistance have been found in several other studies, particularly more recent ones [38,39]. In fact, the widespread use of these antibiotics in the management, treatment, and disease prevention of livestock enhances the selection and development of antimicrobial-resistant *Campylobacter*, which contributes to an increasing burden of antibiotic-resistant infections, with serious consequences for human health due to infection via food chain or direct contact of infected animals.

The majority of isolates have antimicrobial resistance phenotypes that closely match the genetic mutations and genes that code for a corresponding resistance phenotype. The *tet(O)* gene, which produces the TetO ribosomal protection protein, has been linked to tetracycline resistance [40,41]. Our isolates showed a high rate of resistance to fluoroquinolones. The cmeABC operon, encoding for multidrug efflux, was detected in all our isolates irrespective of their resistance or susceptibility to quinolones/fluoroquinolones. The Thr-86-Ile substitution, which is frequently found in isolates resistant to quinolones and fluoroquinolones, is one of several point mutations in the QRDR of the GyrA protein that make up the second resistance process [42]. The gene encoding 23S rRNA had the two-point mutations A2075G and/or A2074C in all erythromycin-resistant *Campylobacter* isolates [42].

The *erm (B)* gene, was found in 55.55% [25/45: 47.61% (10/21) *C. jejuni* and 62.5% (15/24) *C. coli*]. The presence of this gene in *Campylobacter* isolates is noteworthy, as it was discovered on MDR gene islands co-harboring genes that encoded resistance to ampicillin, ciprofloxacin, and tetracycline [43]. These results are concerning because macrolides, in particular erythromycin and azithromycin, are the preferred antibiotics for treating human *Campylobacter* infections. β-lactam antibiotics, such as ampicillin, are intrinsically ineffective against *Campylobacter* spp. [43]. The primary mechanism of acquired ampicillin resistance is, in fact, enzymatic inactivation by the beta-lactamase encoding *bla_OXA-61_* gene, which was found in 20% and 8% of β-lactam resistant *C jejuni* and *C coli* isolates, respectively. According to earlier reports [39,43], the majority of our isolates were gentamicin-susceptible. This may be due to its restricted use for systemic infections and the fact that it is not used in the production of poultry [39,40].

### 3.3. Occurrence of Virulence Genes

The pathogenicity of *Campylobacter* species is influenced by their virulome [44] so it is important to look into the virulence factors of avian *Campylobacter* for consumer safety. 

This study revealed that all isolates had the genes *cdtA, cdtB,* and *cdtC*, which are essential for cytolethal distending toxin (CDT) expression, as well as *flaA*, *cadF*, and *ciaB*, which are related to adhesion, colonization, and invasion. These genes were found in similar frequencies to those previously reported from Korea [45] and Italy [34], but at higher levels than those previously reported from South Africa and Chile [46]. In cell models, *Campylobacter* adhesion and internalization are empowered by the presence of the *cadF* and *ciaB* genes [47]. According to research from South Africa [48], Japan [49], and Iran [50], *C. jejuni* had a higher rate of the *pldA* gene, which encodes the outer membrane phospholipase A. 

## 4. Materials and Methods

### 4.1. Samples Collection

Our study was conducted in an intensive breeding hen farm located in the governorate of Nabeul in north-eastern Tunisia. Eight breeding hen flocks were monitored from May to June 2018. The ages of the breeding hens ranged from 42 to 75 weeks. The farm is one of the biggest poultry farms in Tunisia, encompassing 28 flocks and each flock houses 2300 to 18,000 birds, and covering 43 hectares. Birds had *ad libitum* access to feed and water. The diets were provided in mash form and consisted of corn-soybean meal, barley, bran, calcium carbonate, dicalcium phosphate, and vitamin and mineral supplements. For preventive purposes against infectious diseases, the following antibiotics are commonly used: enrofloxacine, florfenicol, doxycycline, oxytracycline, tiamulin, colistin, and trimethoprim-sulfadiazin. IODOSAN 30 (EWABO Chemikalien GmbH & Co. KG, Wietmarschen, Germany) and DEPTAL MCL (code 02080) (Kersia, Courcelles, Belgium) are used as the disinfectant for cleaning surfaces of flocks.

A total of 362 samples were taken from chickens’ cloacal swabs (*n* = 166) and the farm environment before (*n* = 98) and after (*n* = 98) the cleaning and disinfection of the buildings. Meaning, that before the cleaning period, the 264 collected samples encompassed 166 cloacal swabs and 98 environmental samples (rays, nest boxes, and from the ground). After the cleaning and disinfection step, we collected 98 environmental samples [trays, nests, plates, floor, doors, the changing rooms (blouses, boots, and flip flops)]. Environment samples were taken by smearing swabs soaked in Bolton Broth (Oxoid, Basingstoke, UK). In each flock, approximately 25 cloacal swabs were taken from five locations (four corners and the center).

### 4.2. Audit of Cleaning and Disinfection Procedures in Poultry Farm Buildings

To study the risk factors potentially linked to contamination by *Campylobacter* spp. a questionnaire was properly completed during the sample collection visits for the farm (Appendix A). The farm visit frequency was dependent on the authorizations of the breeders. The epidemiological questionnaire comprised several questions. The questions were related to the description of the farm, the buildings (flocks); breeding characteristics and management; dead bird management measures; control of rodents and other domestic animals; biosecurity; operations personnel and visitors; vaccination and administration of antibiotics. The obtained scores were based on both personal observations and information collected from the employees and the responsible veterinarians. These notes were useful to establish possible correlations between the characteristics of the breeding and the contamination by *Campylobacter* spp. 

### 4.3. Sample Transportation and Processing

According to the Good Execution Analysis (GBEA) Guide and the Standard NF EN ISO 15189 (version 2012) [51], the samples were transported in a cooler box with cold accumulators at 4 °C and brought to the laboratory within 4 h. When the samples arrived at the laboratory, they were all pre-enriched on the same day.

### 4.4. Culture and Growth Conditions 

The isolation of *Campylobacter* was carried out according to the standard French method (AFNOR, La Plaine-saint-denis, France, 1996). All samples were subjected to a selective enrichment step in Bolton broth (Oxoid, Basingstoke, UK), in a microaerophilic atmosphere (5% O_2_, 10% CO_2_, 85% N_2_) using GENbox generators (BioMérieux, Craponne, France) at 42 °C during 18 h–24 h. Presumptive positive culture for *Campylobacter* were streaked on Karmali agar (Oxoid, Basingstoke, UK) and incubated for 48 h under the same conditions as described previously [52]. Under a light microscope and using oxidase/catalase assays, suspected colonies from each sample were examined for typical *Campylobacter* morphology and motility. Then, presumptive *Campylobacter* colonies were subjected to PCR analysis for genus and species identification. 

### 4.5. DNA Extraction

For the PCR tests, the genomic DNA of collected isolates was extracted using the boiling method [39]. *Campylobacter* isolates were grown in 2 mL Bolton broth and plated on Karmali agar. *Campylobacter* colonies were then harvested and suspended in 100 μL TE buffer (10 mM Tris, 1 mM EDTA, pH 8.0). Cell suspensions were heated at 100 °C for 10 min and then cooled to room temperature. Thereafter, cell suspensions were pelleted by centrifugation at 8000 rpm for 5 min. The supernatant containing DNA was collected, transferred into a new tube, and then stored at −20 °C until use.

PCR amplification of a specific fragment of the *16S rDNA* gene using the primers described by Linton et al. [53] was used to *Campylobacter* genus identification of isolates. Then, the isolates were identified as *C. jejuni* or *C. coli* by PCR assays based on amplification of the *mapA* and *ceuE* genes, respectively [54] (Appendix A).

### 4.6. Antimicrobial Susceptibility Test

Antimicrobial susceptibility testing was performed on all isolates using the disk diffusion method on Mueller-Hinton medium (Bio Life, Milan, Italy) according to the recommendation of the Antibiogram Committee of French Society for Microbiology (CA-SFM) [55]. The following antibiotics were used (Oxoid, Basingstocken, UK): ampicillin (AMP,10 μg), amoxicillin/clavulanic acid (AMC, 10/20 μg), tetracycline (TET, 30 μg), erythromycin (ERY, 15 μg), gentamicin (GEN, 10 μg), chloramphenicol (CHL, 30 μg), nalidixic acid (NAL, 30 μg), and ciprofloxacin (CIP, 5 μg) [39]. The isolates were defined as multidrug-resistant (MDR) if they exhibited resistance to at least one agent belonging to three or more antimicrobial families [56].

### 4.7. Detection of Mutation(s) in the QRDR of gyrA and 23S rRNA Genes by PCR Mismatch Amplification Mutation Assay (MAMA-PCR)

A single point mutation Thr-86-Ile in the quinolone resistance-determining region (QRDR) of the *gyrA* gene was defined as the main cause of high-level resistance to quinolones. *Campylobacter* isolates were subjected to analysis by MAMA-PCR, as previously described for *C. jejuni* and *C. coli* isolates by Zirnstein et al. [57,58]. The resistance to macrolide (particularly to erythromycin) is mainly encoded by mutations in the V domain of the *23S rRNA* gene, at the positions A2074C or A2075G. The detection of these mutations was also carried out by MAMA-PCR as described previously by Alonso et al. [59]. 

### 4.8. Detection of Their Resistance Determinants

All *Campylobacter* isolates were screened by PCR to detect the *tet(O)* (tetracyclines resistance), *aphA-3* (aminoglycosides resistance), *cmeB* (multidrug efflux pump), and *bla*_OXA-61_ (β-lactams resistance) genes (Appendix A). The PCR mixture (25 µL) consisted of 2.5 μL of DNA template, 0.2 μM of each primer (Carthagenomics Advanced Technologies, Borj Cédria, Tunisia), 0.2 mM dNTP (Promega, Charbonnières-les-Bains, France), 1X Dream DNA polymerase buffer, and 1U Dream Taq DNA polymerase (Thermo Scientific, Bordeaux, France). 

### 4.9. Detection of Virulence Genes

All *Campylobacter* isolates were tested by PCR for the occurrence of the following virulence genes: *flaA* (motility), *cadF*, *racR*, *virB11*, *pldA, dnaJ* (adherence and colonization), *cdtA*, *cdtB*, *cdtC* (cytotoxin production), *cgtB* and *wlaN* (Guillain-Barré syndrome), *ciaB* (invasiveness), and *ceuE* encoding a lipoprotein in *C. jejuni* strains (Appendix A). 

### 4.10. Data Analysis

All collected data were analyzed using R software. The antimicrobial resistance analyses were performed by means of a Chi-square statistic (*p* < 0.05) [60]. This test is a nonparametric tool designed to compare frequency counts between two groups of different sample sizes; the selection criteria for significantly prevalent variance was a stringent *p*-value of 0.001 or less.

## 5. Conclusions

The present study revealed high levels of *Campylobacter* contamination in the studied breeding farm, with a predominance of *C. jejuni* and *C. coli* species. The recovered strains exhibited high levels of resistance to clinically relevant antimicrobial agents (i.e., erythromycin, ciprofloxacin, and tetracycline) and simultaneously harbored several virulence genes. The persistence of these strains after farm disinfection may present a significant risk of cross-contamination of new chicken batches and the spread of virulent and resistant strains throughout the poultry production chain. It is therefore essential to raise awareness among poultry farmers and to improve biosecurity practices to avoid the persistence and spread of infectious agents, including *Campylobacter*, in poultry farms.

## Figures and Tables

**Figure 1 antibiotics-12-00890-f001:**
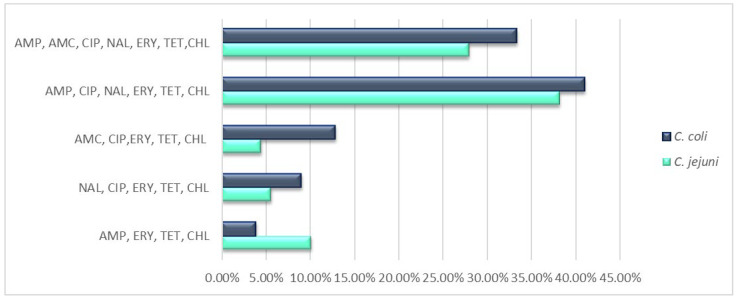
Rates of antimicrobial resistance patterns of *C. jejuni* and *C. coli* isolates. Abbreviations: ERY, erythromycin; CIP, ciprofloxacin; NAL, nalidixic acid; TET, tetracycline; AMP, ampicillin; AMC, amoxicillin/clavulanic acid; CHL, chloramphenicol; GEN, gentamicin.

**Figure 2 antibiotics-12-00890-f002:**
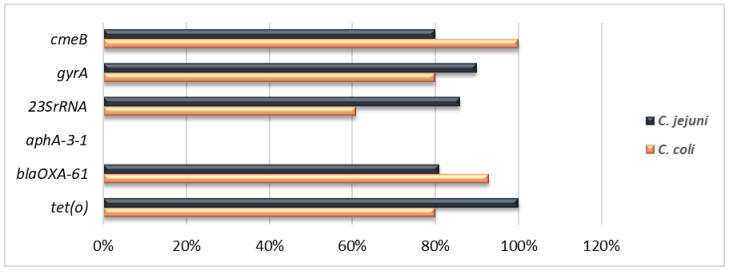
Percentage and distribution of antimicrobial resistance determinants among *C. jejuni* and *C. coli* isolates. Antibiotics and resistance determinants: quinolones (*gyrA*), erythromycin (*23SrRNA*), β-lactams (*bla*_OXA-61_), tetracyclines (*tet*(O)), gentamicin (*aph*A-3), multidrug resistance *(cmeB)*, resistance to ciprofloxacin or nalidixic acid or both and resistance to ampicillin or amoxicillin/clavulanic acid or both.

**Figure 3 antibiotics-12-00890-f003:**
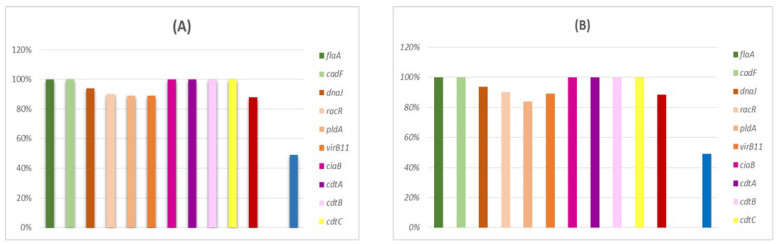
Occurrence of virulence genes in *Campylobacter* isolates (*n* = 167). (**A**) In *C*. *jejuni* (*n* = 78) and (**B**) in *C. coli* (*n* = 89).

**Table 1 antibiotics-12-00890-t001:** Occurrence of *Campylobacter* in environment before and after disinfection.

		Before Disinfection			After Disinfection		
		Nb. Samples	NB % Isolates		Nb.Samples	NB Isolates	
			*C. coli*	*C. jejuni*		*C. coli*	*C. jejuni*
Buildings	Trays	20	0 (0%)	0 (0%)	20	0 (0%)	0 (0%)
	Nestings boxes	20	0 (0%)	0 (0%)	20	0 (0%)	0 (0%)
	Plates	20	12 (60%)	0 (0%)	20	0 (0%)	0 (0%)
	floors	20	20 (100%)	0 (0%)	20	0 (0%)	0 (0%)
Worker’s locker room	Boots	6	6 (100%)	0	6	3 (50%)	0 (0%)
	Slides	6	0	0	6	0 (0%)	0 (0%)
	Blouses	6	0	0	6	0 (0%)	0 (0%)
Total		98	38/98 (38.7%)		98	3/98 (3%)	0 (0%)

**Table 2 antibiotics-12-00890-t002:** Percentage of resistant *Campylobacter* isolates.

Source	Specie	No.	Antimicrobial Resistance Rates % (Number of Isolates)							
			ERY	AM	AMC	Cip	NAL	CHL	Tet	GEN
Layer hens	*C. jejuni*	89	100% (89)	47.19% (42)	58.42% * (52)	100% (89)	100% (89)	100% (89)	100% (89)	1.12% (1)
	*C. coli*	37	100% (37)	81% * (30)	20.22% (18)	100% (37)	100% (37)	100% (37)	100% (37)	0% (0)
Environment	*C. coli*	41	100% (41)	80.48% (33)	21.95% (9)	100% (41)	100% (41)	100% (41)	100% (41)	0% (0)
Total		167	100% (167)	62.87% (105)	47.3% (79)	100% (167)	100% (167)	100% (167)	100% (167)	0.6% (1)

Legends: ERY, erythromycin; AM, ampicillin; AMC, amoxicillin/clavulanic acid; Cip, ciprofloxacin; NAL, nalidixic acid; CHL, chloramphenicol; Tet, tetracycline; GEN, gentamicin. *: significant association between resistance to the corresponding antibiotic and the source of *Campylobacter* isolates.

## Data Availability

The statistical data used to support the findings of this study are available from the corresponding author upon request.

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
