# Peer review of "Persistence of Campylobacter spp. in Poultry Flocks after Disinfection, Virulence, and Antimicrobial Resistance Traits of Recovered Isolates"

_antibiotics, 2023, doi:10.3390/antibiotics12050890_

Round 1

Reviewer 1 Report

The manuscript entitled “Zoonotic Significance of Antibiotic-resistance in Campylobacter in Breeding Hens and Avian Environment in Tunisia: High Helath Risks in the ‘One health’ Approach”, by Manel Gharbi et al., examined the risk factors for Campylobacter contamination in a poultry farm and characterized virulence and antimicrobial resistance determinants of Campylobacter spp. isolated from breeding hens and environmental samples. The Manuscript is well designed and written. I support its publication. Nevertheless, the modifications listed below are required:

Title: I strongly recommend to change the title, as it is really unclear; something like “The zoonotic significance of antimicrobial-resistant Campylobacter species in a poultry farm in Tunisia: a One Health approach” would be much better.

Abstract, line 17: remove comma between “to” and “macrolids”

Abstract, line 19: replace “amoxillin-clavulanic acid” with “amoxicillin-clavulanic acid”

Line 33: foodborne

Line 38: Here and everywhere else in the text, substitute “Campylobacters” with “Campylobacter spp.”

Line 45: remove the comma and replace in with a semicolon (and replace the semicolon in line 46 with a point”

Lines 70-71: the fact that this study was performed in one farm (as I can understand), should be clarified here and in the materials and methods section

Line 110: replace “revealed out to be negative” with “resulted negative”

Line 201: “the survey of poultry farms” it is just one farm right? Please change the sentence

Line 226: the reference “(https://books.google.tn)” is not properly formatted

Line 338: Our study was conducted in an intensive breeding hens farm

Line 345: flip flops

Line 358: replace “Campylobacter” with “Campylobacter spp.”

Line 349: “For each farms” this study was conducted in a single farm of more than one farm?

Line 379: remove the hyperlink from Tris and EDTA

Line 379: at 100°C

Line 429: as there is no supplementary materials, please remove this statement.

Lines 443-459: Data Availability Statement, Acknowledgements and Conflict of Interests sections must be filled or removed.

Line 460: A deep revision of the reference list is required:

Author Response

Reviewer 1

The manuscript entitled “Zoonotic Significance of Antibiotic-resistance in Campylobacter in Breeding Hens and Avian Environment in Tunisia: High Helath Risks in the ‘One health’ Approach”, by Manel Gharbi et al., examined the risk factors for Campylobacter contamination in a poultry farm and characterized virulence and antimicrobial resistance determinants of Campylobacter spp. isolated from breeding hens and environmental samples. The Manuscript is well designed and written. I support its publication. Nevertheless, the modifications listed below are required:

Response: We thank you for your positive evaluation, and we hope this version will meet your appreciation.

  • Title: I strongly recommend to change the title, as it is really unclear; something like “The zoonotic significance of antimicrobial-resistant Campylobacter species in a poultry farm in Tunisia: a One Health approach” would be much better.

Response: Taking in account your comment and those of the other reviewers, we have modified the title as follow: ‘Persistence of Campylobacter spp. in poultry flocks after disinfection, virulence and antimicrobial resistance traits of recovered isolates.

  • Abstract, line 17: remove comma between “to” and “macrolids”

Response: Done

  • Abstract, line 19: replace “amoxillin-clavulanic acid” with “amoxicillin-clavulanic acid”

Response: Done

  • Line 33: foodborne

Response: Done

  • Line 38: Here and everywhere else in the text, substitute “Campylobacters” with “Campylobacter

Response: This was corrected throughout the manuscript.

  • Line 45: remove the comma and replace in with a semicolon (and replace the semicolon in line 46 with a point”

Response: Done

  • Lines 70-71: the fact that this study was performed in one farm (as I can understand), should be clarified here and in the materials and methods section.

Response: The study was conducted on a single farm. It is one of the largest farms in the country, with 28 buildings and 43 hectares. Each building has a rearing capacity of up to 18,000 birds. Samples were taken from only 8 buildings because they contained flocks of the same age and had the same period of sanitation after being emptied, cleaned, and disinfected. The remaining buildings were not sampled because they were still occupied by the flocks at the time of the study. These information were added in the section ‘Materials and Methods’

  • Line 110: replace “revealed out to be negative” with “resulted negative”

Response: Done

  • Line 201: “the survey of poultry farms” it is just one farm right? Please change the sentence

Response: The sentence was modified

  • Line 226: the reference “(https://books.google.tn)” is not properly formatted

Response: this reference is now numbered 28, and added to the list of reference.

  • Line 338: Our study was conducted in an intensive breeding hens farm

Response: This was modified

  • Line 345: flip flops

Response: Done

  • Line 358: replace “Campylobacter” with “Campylobacter

Response: Done

  • Line 349: “For each farms” this study was conducted in a single farm of more than one farm?

Response: It is a single farm, this was corrected

  • Line 379: remove the hyperlink from Tris and EDTA

Response: Done

  • Line 379: at 100°C

Response: Done

  • Line 429: as there is no supplementary materials, please remove this statement.

Response: in the new version, we added supplementary files

  • Lines 443-459: Data Availability Statement, Acknowledgements and Conflict of Interests sections must be filled or removed.

Response: These sections were added.

  • Line 460: A deep revision of the reference list is required:

Response: All references were revised following the instruction of the journal, the name of bacteria are in italic

Reviewer 2 Report

Title: Zoonotic Significance of Antibiotic Resistance in Campylobacter in Breeding Hens and Avian Environment in Tunisia: High Health Risks in the One Health Approach

Major Comments:

Section 4.1: The study lacks information on the sampling criteria, including how the farms were selected and how the sample numbers were determined. Furthermore, the study only sampled from a single location with eight buildings located 20 meters apart, which may not be representative of the entire area. The study should provide a clear sampling methodology to justify its epidemiological approach. Also, the mention of the city area in the title is misleading as the sample was taken from a specific location.

Section 4.2: The authors should mention the name of the disinfectant used at the farm and the contact period of the disinfectant. This information is important in understanding the role of disinfectants in reducing the risk of Campylobacter infection.

• Section 4.6: The study does not provide data on the type of feed given to the breeding hen or the antibiotics used in feed. The authors should gather this data and include it in the study. The introduction mentions the use of antibiotics as growth promoters in the poultry sector, but the study does not discuss the antibiotics used by workers in feed on the sampled farm.

• Line 115: The percentage of the isolates mentioned in line 115 is incorrect. In line 102, the study mentions 89 C. jejuni and 78 C. coli. The prevalence calculations and associated figures (Fig. 3) were calculated based on the wrong database.

• Section 2.1.2: The study lacks data on biosecurity measures followed at the farm. The authors should include this information to provide a complete understanding of the risk factors associated with Campylobacter infection.

• Section 2.2.3: The study mentions the One Health approach in the title but only collected samples from farm workers' clothes and boots. Hand swabs should have been taken to fulfill the criteria of One Health. Furthermore, the study lacks data on hand hygiene, which is essential as workers can act as a source of cross-contamination on the farm.

• Line 217: The methodology mentions eight flocks from eight different buildings, but the article mentions 12.

• The study failed to discuss the One Health approach used to avoid farm contamination. The authors talked about determining the risk factors associated with the infections and preparing questionnaires, but they did not provide any tables or lists based on which the factors were determined. These gaps are significant in the study.

The manuscript is challenging to read and correlate. The results calculated in the study are based on the wrong database, and the authors could not correlate the study with the One Health approach. The title of the study is not justified by the content. The study does not meet the criteria for publication in this journal. In my suggestion, the authors should carefully review their article before submitting it to the journal.

Other Comments:

Line 33: Use 'illnesses' instead of 'illness.'

Line 48-50: Avoid repetition of words.

Line 56: Use italics for Campylobacter.

Line 67, 75: Modify the sentence for fluency.

Line 82: Clarify if it is 2500 and 18000.

Line 98: Use 'were' instead of 'was.'

Table 2: Mention the significance of star marks.

Figure 2: Include color coding of chart, indicating which color represents which organism.

Line 230: No data provided on the worker hygiene. On which criteria the article mentions the given line “Reduction in Campylobacter contamination is achieved when disinfection is carried out by a specialized company” rather than by the farmer himself. Before referencing the line, the author should provide the data from their study.

Line 260: "Our study was conducted between May and June 2018", but in methodology section 4.1, it is mentioned that Eight breeding hen flocks were monitored from October to May 2018. Inconsistences are present in the manuscript. 

Author Response

REVIEWER 2

Major Comments:

  • Section 4.1: The study lacks information on the sampling criteria, including how the farms were selected and how the sample numbers were determined. Furthermore, the study only sampled from a single location with eight buildings located 20 meters apart, which may not be representative of the entire area. The study should provide a clear sampling methodology to justify its epidemiological approach. Also, the mention of the city area in the title is misleading as the sample was taken from a specific location.

Response: We thank you for your insightful comments and for your time in reviewing this manuscript thoroughly. First, we agree with you that the title is confusing. Therefore, in this new version, the title is changed to the following: "Persistence of Campylobacter spp. in poultry flocks after disinfection, virulence, and antimicrobial resistance traits of recovered isolates."

C1: The study lacks information on the sampling criteria, including how the farms were selected and how the sample numbers were determined

Response: As this study is the first in this field in our country, we chose the governorate of Nabeul in northeastern Tunisia, which provides almost 30% of the national poultry production. The selected farm is one of the largest farms in the region. Considering that, the data from this study could be of great interest to evaluate the effectiveness of cleaning and disinfection procedures to avoid cross-contamination between different batches of chickens, as well as further downstream in the poultry production chain, and to study the risk of persistence of virulent strains and the spread of AMR in the microbial populations.

The studied farm is one of the largest poultry farms in the country, with 28 buildings and 43 hectares. Each building has a rearing capacity of up to 18,000 birds. Samples were taken from only 8 buildings because they contained flocks of the same age and had the same period of sanitation after being emptied, cleaned, and disinfected. The remaining buildings were not sampled because they were still occupied by flocks at the time of the study.

C2:“the study only sampled from a single location with eight buildings located 20 meters apart, which may not be representative of the entire area”.

Response: We apologize, the paragraph "Collection of samples" was confusing; indeed, each building is located at 20 m from the others and not only the sampled buildings. The latter were far from each other, and separated by several buildings.

Thus to avoid all these confusions, the paragraph was rewritten as follow:”The study was conducted between May and June 2018, in an intensive breeding hen farm, located in the governorate of Nabeul in northeastern Tunisia in an area of 43 hectares. This farm is one of the largest poultry farms in Tunisia, consisting of 28 buildings, each housing up to 18,000 birds. The age of the hens varies from 42 to 75 weeks. The diet of these birds is based on ad libitum feeding; some antibiotics such as florfenicol or tiamulin may be added occasionally to the diet to prevent the spread of bacterial infections in the flocks. A total of 362 samples were collected from 8 buildings before and after the cleaning and disinfection procedures between poultry batches. Cloacal swabs (n=166) were taken from hens, and environmental samples before (n=98), and after (n=98) building cleaning and disinfection were collected by wiping swabs soaked in Bolton broth (Oxoid, UK). Before cleaning, environmental samples were collected from the combs, nest boxes, and floor. After the cleaning and disinfection, environmental samples were taken from trays, nests, plates, floors, doors, and changing rooms (gowns, boots, and flip-flops). In each flock, the same number of samples were collected at five locations (at the four corners and the center).

Section 4.2: The authors should mention the name of the disinfectant used at the farm and the contact period of the disinfectant. This information is important in understanding the role of disinfectants in reducing the risk of Campylobacter infection.

Response. The used disinfectants were: the IODOSAN 30 (EWABO Chemikalien GmbH & Co. KG) and the DEPTAL MCL (code 02080) (kersia Belgium). The dilution and contact period are applied according to the manufacturer's instructions.

  • Section 4.6: The study does not provide data on the type of feed given to the breeding hen or the antibiotics used in feed. The authors should gather this data and include it in the study. The introduction mentions the use of antibiotics as growth promoters in the poultry sector, but the study does not discuss the antibiotics used by workers in feed on the sampled farm.

Response: You have reason. Theses data are added in this version: “The diet of these birds is based on ad libitum feeding; some antibiotics such as florfenicol or tiamulin may be added occasionally to the diet to prevent the spread of bacterial infections in the flocks.”

  • Line 115: The percentage of the isolates mentioned in line 115 is incorrect. In line 102, the study mentions 89 C. jejuni and 78 C. coli. The prevalence calculations and associated figures (Fig. 3) were calculated based on the wrong database.

Response: We apologize, these mistakes were corrected.  The studied isolates were 89 C. jejuni and 78 C. coli isolates. The statistical analysis was verified and corrected as follows : The rate of Campylobacter detection in the cloacal swabs of breeding hens was 75.9% (126/166), 70.63% of isolates were identified as C. jejuni (n=89) and 29.36% as C. coli (n=37). The percentage of infection in the flocks ranged from 14 % to 56% per building (p < 001).

  • Section 2.1.2: The study lacks data on biosecurity measures followed at the farm. The authors should include this information to provide a complete understanding of the risk factors associated with Campylobacter infection.

Response: A questionnaire was designed to collect data on biosecurity measures from farm managers, veterinarians, and workers, along with observations from the investigator. This questionnaire is added as supplementary file S3 (in French language). Besides, we added this paragraph in “Results “ section to highlight the biosecurity measures applied in this farm ;”According to the biosecurity data obtained through the questionnaire, the biosecurity measures most applied in this farm were the closure of buildings, restriction of visitors, quality of food, administration of required vaccines and application of a disinfection program at the end of each breeding cycle. On the other hand, the least applied measures were the respect of the recommended density of birds per m2, isolation of the farm from external animals (rodents, reptiles, dogs, cats...), use of disinfectant baths, cleanliness of clothing and hand hygiene.

  • Section 2.2.3: The study mentions the One Health approach in the title but only collected samples from farm workers' clothes and boots. Hand swabs should have been taken to fulfill the criteria of One Health. Furthermore, the study lacks data on hand hygiene, which is essential as workers can act as a source of cross-contamination on the farm.

Response: We agree that hand swab samples are lacking, but we collected swabs from staff clothing and boots to investigate their role in the persistence of contamination in the farm environment and the contamination of new batches, thus emphasizing the "One Health" aspect of the study. Nevertheless, we believe that the title is not appropriate, which prompted us to modify it.

  • Line 217: The methodology mentions eight flocks from eight different buildings, but the article mentions 12.

Response: This was corrected

  • The study failed to discuss the One Health approach used to avoid farm contamination. The authors talked about determining the risk factors associated with the infections and preparing questionnaires, but they did not provide any tables or lists based on which the factors were determined. These gaps are significant in the study.

Response: in this version, we focused on the analysis of the factors leading to the persistence of Campylobacter strains in the farm environment, we analyzed also the phenotypic and genotypic characteristics of the recovered isolates to emphasize the risk of cross contamination of the new batches with virulent and MDR strains. These results highlights the risk to consumer health, since these strains could be spread throughout the poultry production chain.

  • The manuscript is challenging to read and correlate. The results calculated in the study are based on the wrong database, and the authors could not correlate the study with the One Health approach. The title of the study is not justified by the content. The study does not meet the criteria for publication in this journal. In my suggestion, the authors should carefully review their article before submitting it to the journal.

Response: the manuscript was revised and modified according to the reviewers' comments, thus we hope this version will meet your satisfaction.

10- Other Comments:

-Line 33: Use 'illnesses' instead of 'illness.'

Response:  This was corrected

-Line 48-50: Avoid repetition of words.

Response: The sentence was shortened and modified as follows: ‘Moreover, the main factors causing the increase of antimicrobial resistance rates in Campylobacter are the over/misuse of antibiotics in human and husbandry.’

-Line 56: Use italics for Campylobacter.

Response: In all the manuscript Campylobacter is written in italic.

-Line 67, 75: Modify the sentence for fluency.

Response: This paragraph was modified as follows: ‘Although contamination of poultry meat is possible at all stages of the food chain, the rearing period represents a critical phase in the spreading of the bacteria.  Therefore, the aim of this study was to investigate the persistence of Campylobacter spp. contamination in one Tunisian intensive breeding hen farm after cleaning and disinfection and to characterize virulence and antimicrobial resistance traits of the recovered Campylobacter strains

-Line 82: Clarify if it is 2500 and 18000.

Response: This was corrected

-Line 98: Use 'were' instead of 'was.'

Response: This was corrected

-Table 2: Mention the significance of star marks.

Response: a legend of this table was added and the ‘star marks’ means significance difference

-Figure 2: Include color coding of chart, indicating which color represents which organism. 

Response: Done.

- Line 230: No data provided on the worker hygiene. On which criteria the article mentions the given line “Reduction in Campylobacter contamination is achieved when disinfection is carried out by a specialized company” rather than by the farmer himself. Before referencing the line, the author should provide the data from their study.

Response: The sentence was deleted to avoid any confusion

Line 260: "Our study was conducted between May and June 2018", but in methodology section 4.1, it is mentioned that Eight breeding hen flocks were monitored from October to May 2018. Inconsistences are present in the manuscript. 

Response: this was correct: the study was conducted between May and June 2018.

Reviewer 3 Report

The introduction can be modified and shortened.

First, the various well-established facts can be deleted without the manuscript losing anything and second, the section can be modified to present the zoonotic significance of the organism and the potential for dissemination of antibiotic resistance to humans.

The objectives of the study must be presented clearly.

4.2. Please include a copy of the questionnaire in supplementary material. How was the questionnaire validated? There are various techniques developed about the accuracy and validity of questionnaire-based work and the authors must present evidence about that, as it is a core-part of their study.

4.5. and throughout the manuscript. For ALL the PCRs the authors MUST give all the details in a supplementary material, not just a reference (primers, annealing temperature, product size etc.) to allow validation of the work performed.

2.1. All the results of the interview through the questionnaire must be present orderly in the supplementary material.

Multi-resistance: there is an official definition of this characteristic and the authors MUST employ this internationally approved term and approach, rather than using arbitrary characterisation.

Discussion

Please add a new sub-section 3.4. with potential one-health implications.

Also, please describe the implications in clinical therapeutics of animals in the farm and more widely.

Overall: extensive improvement and resubmission for further evaluation.

Author Response

Reviewer 3

  • The introduction can be modified and shortened.

First, the various well-established facts can be deleted without the manuscript losing anything and second, the section can be modified to present the zoonotic significance of the organism and the potential for dissemination of antibiotic resistance to humans.

Response: Thank you for your relevant comments and recommendations. According to the comments of other two reviewers, the title of the article was modified as follows: ‘.

The introduction section was modified taking into account your comment, highlighting the zoonotic significance of the organism and the potential dissemination of antibiotic resistant strains to humans.

  • The objectives of the study must be presented clearly.

Response: the aim of the study was modified as follows: Therefore, the aim of this study was to investigate the persistence of Campylobacter spp. contamination in one Tunisian intensive breeding hen farm after cleaning and disinfection and to characterize virulence and antimicrobial resistance traits of the recovered Campylobacter strains.

3- 4.2. Please include a copy of the questionnaire in supplementary material. How was the questionnaire validated? There are various techniques developed about the accuracy and validity of questionnaire-based work and the authors must present evidence about that, as it is a core-part of their study.

Response: As suggested, the questionnaire is added as supplementary material. The used questionnaire was previously prepared and validated by the laboratory LEMV as a national laboratory for avian infectious diseases diagnostics (viral and bacterial infections).

4-,4.5. For ALL the PCRs the authors MUST give all the details in a supplementary material, not just a reference (primers, annealing temperature, product size etc.) to allow validation of the work performed.

Response: The table of primers was provided as supplementary material containing annealing temperature, products sized.

5- 2.1. All the results of the questionnaire must be present orderly in the supplementary material.

Response: As suggested, the results of the questionnaires were included in the supplementary materials.

6- Multi-resistance: there is an official definition of this characteristic and the authors MUST employ this internationally approved term and approach, rather than using arbitrary characterisation.

Response: As suggested we have added the official definition of  multidrug-resistance defined by Magiorakos, et al., (2012), at the end of the section 4.6. Antimicrobial susceptibility test: “The isolates were defined as multidrug-resistant (MDR) if they exhibited resistance to at least one agent belonging to three or more antimicrobial families”.

6- Discussion

Please add a new sub-section 3.4. with potential one-health implications.

Also, please describe the implications in clinical therapeutics of animals in the farm and more widely.

Overall: extensive improvement and resubmission for further evaluation.

Response: according to all comments, the manuscript was modified, we hope this version will meet your satisfaction.

Round 2

Reviewer 3 Report

The authors have missed one or two very recent relevant references with broadly similar work and it will benefit the manuscript if they would include them as well and compare with their results.

Author Response

Dear Reviewer

Thank you for your positive comment on our article. We added 2 articles in order to highlight the phenomenon of Campylobacter persistance after cleaning and disinfection (references 35 and 36) (See lines 292 to 294).

Sincerely